# Osteopathy students profile in Italy: A cross sectional census

**Giacomo Consorti[1], Donatella Bagagiolo[2], Andrea Buscemi[3,4], Luca Cicchitti[5], Michela Persiani [6]\*, Andrea Bergna [7,8], on behalf of OSA Group[¶]**

1 Education Department of Osteopathy, ISO—Istituto Superiore di Osteopatia, Milan, Italy, 2 SSOI—Research Department, SSOI—Scuola Superiore di Osteopatia Italiana, Turin, Italy, 3 CSdOI—Centro Studi di Osteopatia Italiano, Catania, Italy, 4 Horus cooperativa sociale, Ragusa, Italy, 5 AIOT—Accademia Italiana Osteopatia Tradizionale, Pescara, Italy, 6 OSCE—Osteopathic Spine Center Education, Bologna, Italy, 7 SOMA—Istituto Osteopatia Milano, Milan, Italy, 8 AISO—Associazione Italiana Scuole di Osteopatia, Pescara, Italy

¶ Membership of the OSA group is provided in the Acknowledgments.
* ricerca@spine-center.it

**Data Availability Statement:** All relevant data are within the manuscript and its Supporting Information files.

**Funding:** The authors received no specific funding for this work. No The funders had no role in study

## Abstract

### Introduction

The Osteopathy Students Analysis (OSA) aims to profile osteopathy students in Italy as a target population in terms of sociodemographic characteristics, geographical distribution, health status, and previous and ongoing education specifications.

### Materials and methods

The OSA used a cross-sectional design. A Web-based survey was distributed to the Italian Osteopathic Education Institutions (OEIs). The OSA survey was composed of items organised into four sections: 1. Sociodemographic characteristics (11 items); 2. Geographical distribution (5 items); 3. Health status (3 items); 4. Previous and ongoing education specifications (16 items). A descriptive sample population analysis was performed. Dichotomous and categorical variables were presented as frequencies and percentages, and continuous variables were displayed as means and standard deviations. Some variables were analysed using a pentenary distribution.

### Results

49 out of the 61 OEIs identified matched the inclusion criteria, and among these, 22 accepted to propose the enrolment of their students into the study. The survey was administered to 4,720 students from all the participant OEIs. A total of 3,762 students responded to the survey, accounting for an estimated response rate of 53.7%. The majority of respondents were men (54%), with an average age of 26.9 ± 6.5 years. Almost the totality of the sample was composed of the European ethnic group (99.1%). Respondents were predominantly born in Italy (97.2%). The majority of the sample reported being in good (49.5%) to excellent (38.6%) health. To date, osteopathy students are almost evenly distributed between the two types of curricula (T1 = 46.6%; T2 = 53.4%).

design, data collection and analysis, decision to
publish, or preparation of the manuscript.

**Competing interests:** The authors have declared
that no competing interests exist.

**Abbreviations:** AACOM, Association of Colleges of
Osteopathic Medicine; CEN, European Committee
for Standardization; ECTS, European Credit
Transfer System; EQF, European Qualifications
Framework; FORE, Forum for Osteopathic
Regulation in Europe; OEIs, Osteopathic Education
Institutions; DOs, osteopathic physicians; OPERA,
Osteopathic Practitioners Estimates and RAtes;
OSA, Osteopathy Students Analysis; PPE, Peer
Physical Examination; SURGE, SUrvey Reporting
GuidelinE; T1, Type I; T2, Type II; WHO, World
Health Organization.

## Conclusions

The OSA is the first study that aims to profile Italian osteopathy students as a target popula-
tion in terms of sociodemographic characteristics, geographical distribution, health status,
and previous and ongoing education specifications. Future studies should focus on investi-
gating the correlation between the sociodemographic characteristics of students and their
academic performance.

## Introduction

Osteopathy is most widely known as a healthcare approach that offers a system of assessment,
diagnosis and management of many medical conditions with a range of 'hands-on' manual
techniques [1]. Osteopathy has been regulated in many countries, including the US, UK, and
Australia [1]. Nine countries in Europe (Finland, France, Iceland, Denmark, Lichtenstein,
Malta, Portugal, Switzerland and the UK) have legal regulations concerning osteopathy [2–4].
Thus far, despite the regulation process being officially started in Italy, it has not yet been com-
pleted. Hence, the osteopathy courses in Italian universities have not yet been established.
Therefore, osteopathic education is still being carried on in private osteopathic education insti-
tutions (OEIs). Whilst some osteopathic educational models are delivered in universities (e.g.
in the UK, Switzerland, Australia etc), some countries with legal regulations do not share this
(e.g. in France) so regulation does not de facto lead to training delivered in universities in all
the countries. Benchmarks for Training in Osteopathy were formally defined by the World
Health Organization (WHO) in 2010 [5], as a first attempt to make osteopathic education uni-
form worldwide.

However, osteopathic curricula may differ depending on legal and regulatory structures
around the world. Two professional streams in the field of osteopathy have come out amongst
the countries with statutory regulation: 'osteopathic physicians' and 'osteopaths' [1, 6].

Osteopathic physicians (practising osteopathic medicine) are qualified physicians with full
medical practice rights and can specialise in any branch of medical care, while osteopaths
(practising osteopathy) are primary contact healthcare providers with nationally-defined prac-
tise rights who are not licensed to prescribe medications or perform surgery [1, 7].

The more recently published European Committee for Standardization (CEN) Standards of
Osteopathic Healthcare Provision EN 16686 [8] recognises two types of curriculum organisa-
tion proposed in the Benchmarks for Training in Osteopathy. The Type I (T1) programme is
suggested for students without a healthcare background. The T1 programme takes 4,800 hours
to complete over a 5-year period, including at least 1,000 hours of supervised clinical osteo-
pathic practise and training. The Type II (T2) programme is for students who have prior train-
ing as healthcare professionals. The T2 programme takes 2,000 hours to complete over a
minimum of 5 years, including 1,000 hours of supervised osteopathic clinical practise, but this
number might be adapted depending on the range of prior training formats [8]. Nowadays, no
entry exam is required for school access, and only the previous scholar curriculum is evaluated
for T1 or T2 admission.

Worldwide, there has been a growing trend of osteopathic medical school graduates in
recent years. The American Osteopathic Association published an annual snapshot of the
growth and demographics of the profession. In the US, there are 114,425 Doctors of Osteopa-
thy (DOs) and almost 31,000 students enrolled in colleges of osteopathic medicine, as reported
in the 2018 Report on the Osteopathic Medical Profession. These numbers suggest that, if the

upward trend continues, DOs are projected to represent more than 20% of all practising physicians by 2030 [9]. In Italy, there are an estimated range 5,000–10,000 osteopaths, as reported by The Forum for Osteopathic Regulation in Europe (FORE) [3]. Recently, osteopathy has been identified as a health profession in Italy, with Law 3/2018 [10], whereby osteopathic education will move from the private OEIs, as it is now, to the higher education sector under the competence of universities as per Italian law on health care professions education. This decision could change the training scenario of future osteopaths, especially with the introduction of the admission test to universities. The purpose of the admission restrictions is to regulate the number of graduates in the specific programmes in accordance with the expected demand for specific qualifications in the labour market.

In recent years, a great effort has been made, in the scientific production, to profile the osteopathic professionals in many countries [11–16]. One study that aimed to profile osteopathic practitioners in countries without statutory regulations in osteopathy was the Benelux study [13]. Subsequently, the Osteopathic Practitioners Estimates and RAtes (OPERA) census was conducted to profile the osteopathic profession across Europe [12]. Previous osteopathy studies have largely examined educational issues, as to the role of training methods in medical and osteopathic curricula, with particular reference to Peer Physical Examination (PPE) [14], to investigate the attitudes of Italian osteopathic students toward self-directed practise [15] and to assess the student perceived preparedness to practise 1 year after graduation [16].

To the best of our knowledge, there are no studies that take into account the sociodemographic characteristics, geographical distribution, health status, and previous and ongoing education specifications of osteopathy students in Italy. Studying the characteristics of this population represents one of the first steps that may help future research to assess the trend of new professionals licensed yearly. Moreover, data will contribute to the evaluation of the numbers for the limited access to university courses and plan and implement a tailored didactic strategy.

The Osteopathy Students Analysis (OSA) is the first census that aims to profile osteopathy students in Italy as a target population in terms of sociodemographic characteristics, geographical distribution, health status, and previous and ongoing education specifications.

## Materials and methods

### Design

The OSA used a cross-sectional design. The study was written in accordance with the SUrvey Reporting GuidelinE (SURGE) [17].

The study protocol was approved by the Ethics Committee of the University of Catania Department of Education Science (29 Dec 2018).

### Population

In this survey, the reference population included students attending the Italian OEIs. In order to identify the Italian OEIs, a search strategy was established through a web search using the following keywords: "osteopathic school", "osteopathic university", "osteopathic academy", "osteopathic institute", "osteopathic education", "osteopathic profession", "osteopathic course", "osteopathic degree", "osteopathic bachelor of science", "osteopathic master of science", and "osteopathic diploma".

The web search was performed independently by two of the authors in May 2019.

Once retrieved, the Italian OEIs were contacted by a member of the team of researchers via e-mail in order to obtain information on the educational level, when not presented in the web site. In this study, the EN16686 standard criteria were adopted, due to the fact that it sets a

benchmark for high-quality clinical practise, education, safety and ethics. These criteria help osteopaths to ensure that they provide high-quality healthcare services [4, 5, 8].

The Italian OEIs were eligible for the study if they fulfilled at least one of the EN16686 criteria: as many as 4,800 hours of education, as many as 240 European Credit Transfer System (ECTS) with at least 60 ECTS of European Qualifications Framework (EQF) level 7 (EQF level 7 degree). The Italian OEIs that organised only postgraduate courses in osteopathy were excluded. The sample population consisted of Italian OEIs students, both in T1 and T2 training programmes, who were enrolled in the 2019–2020 academic year, in each year of the course to ensure better sample representativeness [18]. Before receiving an online questionnaire survey, the Italian OEIs that met the inclusion criteria signed the consent form in electronic format.

## Survey questionnaire

The OSA survey was composed of closed ended items organised into four sections: 1. Sociodemographic characteristics (11 items); 2. Geographical distribution (5 items); 3. Health status (3 items); and 4. Previous and ongoing education specifications (16 items). A detailed overview of the items and the respective answer categories is provided in S1 File. The items were selected through a critical analysis of the relevant literature in order to define the specific items of each section [12,19–21]. Personal background features, such as demographic profile, health status and previous education, are reported as the basic knowledge that contributes to the student's performance and knowledge acquisition evaluation [22]. A preliminary pilot-test was administered to 30 Italian graduate students, with a profile similar to that of the sample population, to test the comprehensibility, relevance, acceptability, feasibility and completeness of the survey information [23]. When the questionnaire was completed, the interpretation of the items was gathered through an interview with the respondents [24]. The researchers reviewed some of the unclear items to make the survey more comprehensible.

The survey was promoted in the Italian Association Schools of Osteopathy website (www. aiso-associazionescuoleosteopatia.it) and was performed through dedicated online survey software, which was accessible both from mobile devices and computers.

One researcher panel was in charge of collecting and accessing the anonymised dataset. The study respected the anonymity and privacy of data in accordance with the General Data Protection Regulation (GDPR Regulation EU 2016/679) [25]. Answers were anonymised and IP addresses were not visible to the researcher. The system automatically managed the link between study ID, e-mail address, and survey status; therefore, it was not possible to identify the respondents.

## Administration

The survey was conducted through the performance of a census from 1 October 2019 to 31 January 2020. Each of the Italian OEIs was responsible for administering the survey to all of their students by e-mail with a link to the survey. Participation in the survey was voluntary and anonymous. Consent to participate was requested before the administration of the survey and was given in electronic format. They had to understand and accept all the information reported in the introduction page of the online survey. In order to promote the survey, students were also encouraged to actively take part in it by the personal involvement of researchers and class representatives.

## Statistical analysis

The sample size was calculated based on information obtained through the web and Italian Association of Schools of Osteopathy, which supported an estimated sample of 7,000 students.

According to the literature, it was assumed that the response rate ranged between 50% to 60% of those receiving the questionnaire [12]. In our survey we supposed that the percentage of non-respondents was 50% of OEIs students, so at 95% of the confidence level with a 5% margin of error, the percentage of the students was expected to range between 45% and 55% with a number ranging from 3,150 to 3,850 [26].

A descriptive sample population analysis was performed. Dichotomous and categorical variables were presented as frequencies and percentages, and continuous variables were displayed as means and standard deviations. Some variables were analysed using a pentenary distribution [27].

## Results

The screening individuated 61 OEIs. In all, 49 OEIs matched the inclusion criteria, and among these, 22 accepted to propose the enrolment of their students into the study. The survey was administered to 4,720 students from all the participant OEIs. A total of 3,762 students responded to the survey, accounting for an estimated response rate of 53.7%.

### Sociodemographic

The majority of respondents were men (54%), with an average age of 26.9 ± 6.5 years. Almost the totality of the sample was composed of the European ethnic group (99.1%), and more than half of the respondents declared to be Christian (63.6%) (Table 1). The majority of students declared to be a working student (75.9%), and the most represented profession category among them was "self-employed" (35.2%). The scholarship of both parents of the respondents showed similarities, accounting for a majority with a high school diploma (mothers: 45.4%;

Table 1. Socio-demographic characteristics.

| Parameter | N | % |
|---|---|---|
| Total | 3762 | 100 |
| **Gender** | | |
| Woman | 1721 | 45.7 |
| Man | 2030 | 54.0 |
| Other | 11 | 0.3 |
| **Ethnic Group** | | |
| European | 3727 | 99.10 |
| Arab | 6 | 0.16 |
| Indian | 2 | 0.05 |
| Mixed | 11 | 0.30 |
| Black | 3 | 0.08 |
| Central Asian | 3 | 0.08 |
| American Indian | 1 | 0.03 |
| East Asian | 4 | 0.08 |
| Inuit | 2 | 0.04 |
| Mestizo | 3 | 0.08 |
| **Religion** | | |
| Christian | 2391 | 63.6 |
| Atheist | 1012 | 26.9 |
| Muslim | 8 | 0.2 |
| Jew | 4 | 0.1 |
| Other | 347 | 9.2 |

fathers: 46%). Similarly, the majority of parents were in fulltime employment (56.7% of mothers and 57.2% of fathers) (Table 2). A wide portion of the sample declared to engage in sports activities (76.2%).

## Geographical distribution

Respondents were predominantly born in Italy (97.2%), and every Italian region was represented in the sample, ranging from 10 (Valle d'Aosta) to 966 (Lombardia) participants. Despite the geographical provenience being distributed among all the Italian regions, 39.9% of respondents declared to study offsite from their place of residence. As a result, the regions where the respondents declared to study showed a different distribution (Table 3). Moreover, 86.9% of respondents exposed they will work in their region of origin.

## Health status

The majority of the sample reported being in good (49.5%) to excellent (38.6%) health. However, 7.1% of the respondents declared that they suffered from musculoskeletal disorders. Only 14 respondents (0.4%) had a severe condition that allowed them to access a disability pension

**Table 2. Family of origin characteristics.**

| Parameter | N | % |
|---|---|---|
| **Mother education level** | | |
| Elementary school | 114 | 3 |
| Secondary school | 630 | 16.8 |
| High school | 1709 | 45.4 |
| University | 1255 | 33.4 |
| PhD | 54 | 1.4 |
| **Father education level** | | |
| Elementary school | 165 | 4.5 |
| Secondary school | 851 | 22.6 |
| High school | 1731 | 46 |
| University | 946 | 25.1 |
| PhD | 69 | 1.8 |
| **Mother occupation** | | |
| Self-employed | 693 | 18.5 |
| Fixed-term employee | 263 | 6.9 |
| Long-life employee | 2133 | 56.7 |
| Household | 673 | 17.9 |
| **Father occupation** | | |
| Self-employed | 1415 | 37.6 |
| Fixed-term employee | 163 | 4.3 |
| Long-life employee | 2153 | 57.2 |
| Household | 31 | 0.9 |
| **Yearly income** | | |
| 0–15.000 € | 222 | 5.9 |
| 15.001–28.000 € | 678 | 18 |
| 28.001–55.000 € | 809 | 21.5 |
| 55.000–75.000 € | 310 | 8.2 |
| >75.000 € | 213 | 5.7 |
| Not declared | 1530 | 40.7 |

**Table 3. Geographical distribution.**

| Region | Region of birth | | Region of residence | | Region of education | |
|---|---|---|---|---|---|---|
| | N | % | N | % | N | % |
| Abruzzo | 146 | 3.81 | 161 | 4.32 | 263 | 7.06 |
| Basilicata | 22 | 0.59 | 20 | 0.54 | 1 | 0.03 |
| Calabria | 47 | 1.15 | 37 | 0.99 | 5 | 0.13 |
| Campania | 143 | 3.74 | 121 | 3.25 | 78 | 2.09 |
| Emilia | 367 | 9.75 | 394 | 10.57 | 487 | 13.07 |
| Friuli | 50 | 1.34 | 43 | 1.15 | 37 | 0.78 |
| Lazio | 724 | 19.43 | 777 | 20.85 | 921 | 24.72 |
| Liguria | 30 | 0.81 | 28 | 0.75 | 8 | 0.21 |
| Lombardia | 966 | 25.93 | 1015 | 27.24 | 1179 | 31.64 |
| Marche | 138 | 3.70 | 141 | 3.78 | 71 | 1.80 |
| Molise | 16 | 0.43 | 18 | 0.48 | 5 | 0.13 |
| Piemonte | 218 | 5.74 | 228 | 6.12 | 200 | 5.16 |
| Puglia | 255 | 6.74 | 221 | 5.93 | 152 | 4.08 |
| Sardegna | 24 | 0.64 | 18 | 0.48 | 3 | 0.08 |
| Sicilia | 183 | 4.81 | 175 | 4.70 | 137 | 3.57 |
| Toscana | 104 | 2.78 | 108 | 2.90 | 71 | 1.80 |
| Trentino | 17 | 0.46 | 17 | 0.46 | 1 | 0.03 |
| Umbria | 34 | 0.81 | 27 | 0.72 | 8 | 0.11 |
| Valle d'Aosta | 10 | 0.27 | 12 | 0.32 | 2 | 0.05 |
| Veneto | 176 | 4.61 | 191 | 5.13 | 128 | 3.33 |
| Foreign | 92 | 2.46 | 10 | 0.27 | 5 | 0.13 |

(Table 4). Noticeably, 244 students, accounting for 6.5% of the participants, declared to suffer from "other" health issues.

## Previous and ongoing education

Osteopathy student respondents chose the osteopathy course mostly for the professionalising nature of osteopathic courses in Italy (63.2%) and for the interest in the study topics (60.6%). The choice to engage in the osteopathic school was autonomous for 82.7% of the respondents, and the driving factor for the school choice was mostly the quality of the teachers (60.8%). To date, osteopathy students are almost evenly distributed between the two types of curricula (T1 = 46.6%; T2 = 53.4%). The most represented high school background for the respondents was the scientific high school (50.3%). Although 15.9% of the sample declared to have failed at least once an academic year during high-school, the graduation grades of the sample were distributed among the 1st and the 2nd pentiles [27]. More than half of the sample had a higher education degree. Namely, 53.2% had a BSc, 9.3% had a MSc and only 0.6% already had a PhD. Despite the fact that 36.9% of respondents did not have an academic degree, only 27.5% of them had never attended a university. The average graduation grade for respondents with a higher education degree fell into the 1st pentile (103.7/110 ± 8.4). To date, only a few of the osteopathy students who answered the questionnaire declared to have decided to relocate to another osteopathic school once (4.8%) or twice (0.3%). The distribution of the students among the years of the curriculum was similar among the years (16.8% to 19.6%), apart from the 6th year (10.3%). The year fail rate among the entire curriculum was 3.2% (Table 5). Supplementary Tables with data not shown in the manuscript are provided in S1 Table.

**Table 4. Health status.**

| Parameter | N | % |
|---|---:|---:|
| **Health status perception** | | |
| Very poor | 7 | 0.2 |
| Poor | 45 | 1.3 |
| Fair | 392 | 10.4 |
| Good | 1864 | 49.5 |
| Excellent | 1454 | 38.6 |
| **Impaired system** | | |
| Central Nervous System | 40 | 1.1 |
| Peripheral Nervous System | 21 | 0.6 |
| Sensorial System | 22 | 0.6 |
| Musculoskeletal System | 267 | 7.1 |
| Cardiovascular System | 61 | 1.6 |
| Endocrine System | 139 | 3.7 |
| Respiratory System | 88 | 2.3 |
| None | 2996 | 79.6 |
| Other | 244 | 6.5 |

## Discussion

The OSA survey research provides the first large-scale nationally-representative analyses of the sociodemographic characteristics, geographical distribution, health status, and previous and ongoing education specifications of osteopathy students in Italy. Our sample was reasonably representative of Italian osteopathy students, as we had a 53.7% response rate and the participating OEIs were distributed throughout Italy. Data could be considered sufficient to generalize a number of interesting findings concerning the profile of osteopathy students in Italy.

Our results showed a gender distribution comparable with the OPERA-IT study [12]. In fact, the OPERA-IT study showed that, in the newer generation of osteopaths, the women/men ratio tends to be even. The osteopathic profession in Italy has undergone progressive

**Table 5. Education.**

| Parameter | N | % |
|---|---:|---:|
| **Curriculum** | | |
| T1 | 1754 | 46.6 |
| T2 | 2008 | 53.4 |
| **Academic title** | | |
| BSc | 2000 | 53.2 |
| MSc | 349 | 9.3 |
| PhD | 21 | 0.6 |
| None | 1392 | 36.9 |
| **Actual academic year** | | |
| 1 | 721 | 19.2 |
| 2 | 738 | 19.5 |
| 3 | 632 | 16.8 |
| 4 | 646 | 17.2 |
| 5 | 639 | 17 |
| 6 | 386 | 10.3 |

feminisation since its inception [12]. This data seems to be more evident in a European study [16] in which the study sample was predominantly composed of female students (73.8%). Despite the fact that the authors declared that the selected sample might have not been representative of the population of European osteopathy students, it might be speculated that, if the same trend applied to other countries in which the osteopathic profession has been established prior to than in Italy (e.g., the UK), the observed ratio might be plausible.

With respect to religious creed, the vast majority of the participant students (73.1%) claimed to be religious and 26.9% to be atheists. These data are in line with those collected last year for the new generation of Italians between 15 and 34 years of age, which would seem to be less religious than the previous ones: Catholics just over 50% and atheists/agnostics over 22%, with Catholics increasing with age to reach a peak of 76.9% among those over 50 [28]. Caring for others can be one of the fundamental aptitudes for the healthcare professions and therefore also for osteopathy. In fact, a physician's skills include patient-centred care, promoting well-being, responding to the needs of patients and society, and sensitivity to the patient's culture [29]. In particular, osteopathy supports "whole-person care" and the "holistic approach of healthcare", considering the body-mind-spirit paradigm as a fundamental principle of its treatment proposal [30]. The mind-body-spirit medicines combine characteristics of ancient traditional healing systems with the modern biomedical model to create an integrated approach to healthcare [31]. The Christian community provides an alternative nurturing context that permits the full development of the healing relationship and its essential elements: the fact of illness, the profession of healing, and the act of medicine [32]. For example, nursing students state they chose the profession due to altruistic motives of caring for others, and it would seem that Christian students especially have this perspective [33]. However, an excessive polarization towards a single religion (as well as towards every cultural trait) might expose students to a lack of "diversity" awareness, resulting in a lack of multicultural care competence. Therefore, it might be needed to incorporate multicultural care training into the osteopathic curriculum to allow students to help patients from all faiths/spiritual backgrounds. Since an educational foundation on spiritual care is integral for the healthcare professions [34, 35], the religious characteristics of Italian culture, also present in undergraduate programmes, could favour the choice of care professions. The same students of some healthcare professions, such as nursing sciences, affirm the importance of spiritual awareness in order to address the spiritual needs of patients, incorporating spirituality into education and practice: "Without spirituality, a person is not considered whole" and "Spirituality is an important aspect of a human being" [36–38]. Therefore, spirituality could favour the acquisition of useful skills in the healthcare professions and seems to not negatively influence the acceptance of osteopathy students in Italy in the engagement of teaching/learning activities often used, such as technical and manual skill peer to peer practice) [14], where the students act as models for each other in learning skills in physical examination. Moreover, it has also been claimed that students will be more sensitive and humane and have greater empathy for their patients with this personal experience using PPE [39].

The survey showed that students chose to study osteopathy independently (82.7%). But, in the absence of a previous academic qualification (36.9%), the choice is secondary to a first test of access to a university education (9.4%). However, osteopathic training is sought after a previous academic title. In fact, more than half of the students surveyed already had a degree. Training in osteopathy is rarely considered after a PhD (0.65%), which is probably due to the absence of an academic career. Currently, in Italy, the osteopathic profession gives the possibility of working only as a self-employed; 35% of students declare to be self-employed, and this could be justified by a request in specific training in manual therapy to increase work performance [40] and by the demand of patients who increasingly resort to non-pharmacological therapies [41]. Osteopathy has a large scope, from premature newborns [42, 43] to pregnant women [44] to the elderlies [45]. It is widely used in sports both to improve performance [46],

to prevent and reduce injuries [47–49] and for musculoskeletal disorders [44, 50]. The multi-disciplinary nature in which osteopathy can be applied makes the osteopath a versatile figure with wide working prospects both alone and in a team. The current structure of training in Italy (T1 and T2) allows a comparison in class, mixed classes, between non-working students, workers and teachers, reducing the level of fatigue and increasing the degree of satisfaction of students [51]. It could also be a good teaching strategy to be deepened with further studies and to reduce stress, anxiety and depression, which are very frequent symptoms among students (depression 51.3%, anxiety 66.9%, stress 53%) [52]. In order to assess the outcome of the curriculum in future studies, it would be useful to evaluate how many graduates in osteopathy work exclusively as osteopaths. Similarly, it would be useful to assess whether osteopaths without previous healthcare qualifications (T1) decided to acquire an academic qualification after the osteopathic curriculum. This would allow us to understand how osteopathy is placed in the Italian working and cultural system.

Concerning the relationship between students' educational choice and education level of their parents, the findings of this study are in accordance with national trends [53]. Most osteopathy students in Italy have parents with a high school diploma (46%), followed by a university degree (29%). Moreover, nearly 60% of the respondents have both parents with long-life employment, with an annual income of € 18,000 and € 55,000, which is also in line with national standards [53]. Considering that the cultural, occupational and social backgrounds of the parents have a decisive influence on the educational choice of the respective grown children [54, 55], it might be possible to hypothesise that the osteopathic profession is considered a consistent profession with a high standard of living.

Interpreting the geographical distribution through the European nomenclature of territorial units for statistics [56], it is possible to compare the provenience of our participants and the geographical distribution of professional osteopaths presented in the OPERA-IT study [12]. Our data support the geographical distribution observed in the OPERA-IT study among professional osteopaths. Indeed, the North-West macroregion was the most common area of provenience (n = 12,24), followed by the Centre (n = 1,146), North-East (n = 610), South (n = 483) and Islands (n = 207) macroregions.

The students in the assessed sample stated that they had good and excellent health status, with a percentage of 88.1%. According to data provided by the Health Behaviour in School-Aged Children study conducted in Italy in 2010 [57], in which the percentage of young Italians who considered themselves in good or very good health was above 85%. With regards to the 6.5% of students reporting"other" health issues, it is well known that anxiety, depression and burnout are frequent conditions among students. Since the answer "other" was closed ended it is not possible to determine to what extent respondents included psychological disorders into that category. It would be interesting for future studies to further investigate the prevalence of psychological conditions among osteopathy students [52, 58, 59].

Moreover, it is in line with a study on university students from 23 European countries, with differences in culture, in which 87.7% and 90.4% of students of Central-Eastern and Western European countries, respectively, evaluated their own health to be at least "good" [60]. Based on the WHO definition of health [60], it is necessary to consider not only the physical well-being but also the social and mental well-being in order to have a complete understanding of the health of individuals. For this reason, self-perceived health is closely related to multifactorial well-being. Indeed, life satisfaction has been shown to be a strong predictor of self-perceived health [61] and to be associated with health-promoting behaviours, such as not smoking, physical exercise and limiting fat intake [62]. The self-perceived good health of osteopathy students in Italy could therefore be an appropriate attitude to the skills of the osteopath, which include promoting well-being and responding to the needs of the patients [63].

The analysis of Italian osteopathy students is interesting considering the growing interest of high school graduates in the faculties in the health professions [64]. The data of the Italian Ministry of Education, University and Research shows that, for the academic year 2018–2019, the faculties of Medicine and Social Healthcare are among the first choices of those who enrol at universities (32,000 students), along with those in Economics, Engineering and Science. A trend that could be anything but casual and that would be correlated to the labour market. Those who graduate in these subjects, in fact, work and earn more than the other graduates 5 years after obtaining the title, as reported in the latest surveys by the AlmaLaurea Inter-University Consortium (employment rate = 89.3%, with an average net salary 5 years after graduation of 1503.00 euros, data relating to graduates in 2013).

The number of osteopathy students remains among the fastest growing healthcare education choices in some countries and shows no signs of slowing down. The American Association of Colleges of Osteopathic Medicine (AACOM) reported 21,584 osteopathic medical school applications in 2019 (+ 5.8% applications compared to 2018) [65]. Similar growth has been registered by the Australian osteopathic workforce, with 2,627 osteopaths in 2019, compared to 1,601 registered practising osteopaths in 2012 [66]. In the UK, there is a similar trend in terms of numbers of students. The General Osteopathic Council published, on 1 March 2019, that there were 5,341 osteopaths on the UK Statutory Register of Osteopaths compared to 4,688 registered practicing osteopaths in 2012 [67, 68]. However, Australian and UK data refers to professionals. Therefore it has to be considered a delay of, at least, 4 years from the time of their enrolment into the osteopathic school. Although in the US, Australia and the UK, the majority of students are represented by women (the US 11,347, Australia 1,422, and the UK 2,710) [66–70]. In Italy, the male population is still the most represented (2,030), as reported in the present work. However, albeit slowly, this trend is aligning with other countries. Osteopathic students are younger in the US and Australia, with a mean age of 24 years and 25 years, respectively, compared to in the UK (mean range 31–50 years) and Italy (mean age 29 years) [65–69]. This civil registry difference is given by a differentially designed educational system between countries. So far, only the AACOM has published an exhaustive report on student characteristics. They showed that, although the percentage of under-represented minorities increased by 0.6% (from 16.4% in 2018 to 17% in 2019), Whites and Asians continue to make up more than 75% of the applicant pool. The majority of US applicants have obtained a baccalaureate education degree (life sciences 74.3%, social sciences 9.9%, physical sciences 6.2%, and arts and humanities 5.1%), although most of them came from economically disadvantaged households and were living in a large town [65]. On the other hand, ethnic minorities are under-represented among osteopathy students in Italy. As reported in this paper, almost the totality of the sample was composed of the European ethnic group (3,727). As is such for US students, the majority of Italian osteopathic applicants have obtained baccalaureate education degrees, although most of them came from families with low to middle income bracket households and were living in a large town. The worldwide picture of osteopathic students is almost uniform, suggesting that more information is needed to better understand the future trend of this population in order to assist policy-makers and educational institutions to tailor their offer to this specific target.

## Limitations

Our survey research has certain limitations. Firstly, the web-based search strategy to detect the Italian OEIs may have missed some of the institutions that could have participated in the survey, and this could underestimate the population size.

The response rate reached in the survey was considered sufficient. However, some literature suggested that the response rate should be 70–80% [70] for a more confident generalisation of

the findings to the entire population. The lower the response rate is, the greater the probability that those who answered would be self-selected rather than randomly chosen, since it is not always possible to determine why other subjects did not respond. Another problem is the potential for nonresponse bias, the probability of which decreases as response rate increases [71].

In the literature, it is widely demonstrated how the occupational field of parents affects the child's choice of educational fields [72]; moreover, choosing the same educational field as one's parents helps the grown children to reach a similar social and occupational status [54]. For these reasons, one of the limitations of this study is the lack of information related to the occupational domain of the parents.

## Conclusions

The OSA is the first study that aims to profile Italian osteopathy students as a target population in terms of sociodemographic characteristics, geographical distribution, health status, and previous and ongoing education specifications. Our findings have been built upon an estimated representative sample. The outlined profile shows that Italian osteopathy students are evenly distributed between gender, mostly belonging to European ethnic group, and religion, mostly Christians. The educational levels and occupations of the parents of students are comparable, accounting for a high school diploma and long-life employee, respectively. The North-West macroregion accounts for the highest number of students followed by the Centre, North-East, South and Islands macroregions. Overall, students declared to have a good health status with the musculoskeletal system appearing to be the most impaired system. The majority of students are engaged in a T2 curriculum.

Future studies should focus on investigating the correlation between the sociodemographic characteristics of students and their academic performance.

## Supporting information

**S1 File. Questionnaire.**
(DOCX)

**S2 File.**
(DS_STORE)

**S1 Table. Supplementary tables.**
(DOCX)

## Acknowledgments

The authors thank OSA Group which is composed by: Marcello Marasco (AbeOS), Nunzia Esposito (AEMO), Stefano Polistina (AIFROMM), Marco Giardino, Alberto Maggiani, Roberto Scognamiglio, Ferdinando Zucchi (AIMO), Gianfranco Pizzolorusso (AIOT Marche), Luca Cicchitti (AIOT Pescara), Vito Adragna (AISERCO), Gianfranco Pizzolorusso (ATSAI), Marco Petracca (CERDO), Andrea Graffitti, Carmine Tuoto (Chinesis IFOP), Luca Lombardi (CIO Collegio Italiano Osteopatia), Andrea Buscemi (CSdOI), Vito Adragna (CSOT), Giulio Tedesco (ICOMM), Carmine Castagna, Viviana Pisa, Stefano Uberti (ISO), Emanuele Chiggiato (FULCRO), Roberto Pagliaro (Osteopathic College Trieste), Michela Persiani (OSCE Bologna), Marta Simonelli (SOFI), Andrea Bergna (SOMA Istituto Osteopatia Milano), Donatella Bagagiolo (SSOI Scuola Superiore di Osteopatia Italiana Torino), Gianpaolo Tornatore

(TCIO). They also thank Marinella Coco (Department of Biomedical and Biotechnological Sciences, University of Catania, Catania, Italy).

## Author Contributions

**Data curation:** Donatella Bagagiolo, Andrea Buscemi, Luca Cicchitti, Michela Persiani, Andrea Bergna.

**Formal analysis:** Giacomo Consorti.

**Methodology:** Giacomo Consorti, Donatella Bagagiolo, Andrea Buscemi, Luca Cicchitti, Michela Persiani, Andrea Bergna.

**Project administration:** Andrea Bergna.

**Supervision:** Andrea Bergna.

**Writing – original draft:** Giacomo Consorti, Donatella Bagagiolo, Andrea Buscemi, Luca Cicchitti, Michela Persiani, Andrea Bergna.

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
