## [Decision Letter · Decision Letter 0]

22 Dec 2020

PONE-D-20-33047

Italian Osteopathy Students Profile: A Cross Sectional Census

PLOS ONE

Dear Dr. Persiani,

Thank you for submitting your manuscript to PLOS ONE. After careful consideration, we feel that it has merit but does not fully meet PLOS ONE’s publication criteria as it currently stands. Therefore, we invite you to submit a revised version of the manuscript that addresses the points raised during the review process.

We look forward to receiving your revised manuscript.

Kind regards,

Jenny Wilkinson, PhD

Academic Editor

PLOS ONE

Journal Requirements:

2.  Please change "female” or "male" to "woman” or "man" as appropriate, when used as a noun (see for instance https://apastyle.apa.org/style-grammar-guidelines/bias-free-language/gender).

3. Please note that according to our submission guidelines (http://journals.plos.org/plosone/s/submission-guidelines), outmoded terms and potentially stigmatizing labels should be changed to more current, acceptable terminology. For example: “Caucasian” or "European Caucasoid" should be changed to “white” or “of [Western] European descent” (as appropriate).

4. One of the noted authors is a group; OSA Group.

In addition to naming the author group, please list the individual authors and affiliations within this group in the acknowledgments section of your manuscript.

Please also indicate clearly a lead author for this group along with a contact email address.

5. Please include captions for your Supporting Information files at the end of your manuscript, and update any in-text citations to match accordingly. Please see our Supporting Information guidelines for more information: http://journals.plos.org/plosone/s/supporting-information

Additional Editor Comments:

Thank you for your submission. Reviewers have provided detailed comments which are provided for your consideration and response.

Reviewers' comments:

Reviewer's Responses to Questions

**Comments to the Author**

1. Is the manuscript technically sound, and do the data support the conclusions?

Reviewer #1: Yes

Reviewer #2: Yes

2. Has the statistical analysis been performed appropriately and rigorously? 

Reviewer #1: Yes

Reviewer #2: Yes

3. Have the authors made all data underlying the findings in their manuscript fully available?

Reviewer #1: Yes

Reviewer #2: Yes

4. Is the manuscript presented in an intelligible fashion and written in standard English?

Reviewer #1: Yes

Reviewer #2: Yes

5. Review Comments to the Author

Reviewer #1: Thank you for the opportunity to review this interesting and important cross-sectional survey of Italian osteopathic students.

This is a clear and well-presented manuscript and will enhance the evidence base regarding osteopathic educational characteristics. I have made some recommendations for amendments which may enhance clarity particularly regarding phrasing for an English language submission.

In table two the term mulatto is used, this is considered to be a derogatory term in English and consideration should be given to replacing it with “Mixed race” or similar.

The sentence Similarly, the working position of both parents accounted for a majority of permanent job positions (mothers: 56.7%; fathers: 57.2%) (Table 2).” Is somewhat unclear could this be presented as an amalgamated % of employment and then described or reworded “ The majority of parents were in fulltime employment (%6.7% of mothers and 57.3% of Fathers) . The information in table two is not aligned in Table 2 this causes some issues of clarity.

In the limitations section “helps the grown sons to reach a similar social and occupational status” should be reworded to a gender neutral term.

Thank you for the opportunity to review this work

Reviewer #2: Thank you for the opportunity to review this manuscript. The survey assessed students currently training in osteopathy in Italy. A newly designed questionnaire was developed to profile this population. I have made comments in the manuscript directly regarding changes to consider to help with clarity for the readers.

Overall the manuscript is well written and the methods are appropriate for the overall aim.

On the top of the comments in the PDF attached, I have more general comments I would like to offer here:

- the authors declare no conflict of interests but some of them are affiliated with some of the listed Italian schools: could this have influenced how data was handled? could this have influenced the choice for some institutions to take part or not (a third of all institutions fitted the inclusion criteria)?

- I would suggest considering changing the title and description in the manuscript of the population assessed: they are referred as Italian students but it may be more accurate to describe them as students in Italy. First because some Italian study abroad (e.g. in the UK) and secondly because you may have non-Italian students in your sample (although this is unclear as I don't think you report the answer to question 12 of the questionnaire regarding their nationality)

- some of the questionnaire questions have "other" as a possible answer. Did they lead to an open-ended box where they could enter some text? If so how were the data handled? This is not detailed in the methods or results sections clearly.

- how was the questionnaire set: could participants select several answers or only one? e.g. for question 24 on school choice (and the results on that question are not presented in a table - if too many tables are required maybe they could be added in supplementary material?)

6. PLOS authors have the option to publish the peer review history of their article (what does this mean?). If published, this will include your full peer review and any attached files.

Reviewer #1: No

Reviewer #2: **Yes: **Dr Jerry Draper-Rodi

---

## [Author Response · Author response to Decision Letter 0]

2 Feb 2021

Manuscript number PONE-D-20-33047, entitled “Italian Osteopathy Students Profile: A Cross Sectional Census”

Dear editor,

Dear reviewers,

We greatly appreciate your readiness to have read our paper and to provide us with relevant feedback and useful suggestions to further improve the quality of our paper. A detailed description of all changes has been provided below.

EDITOR

Response: Thank you for your comment. Done.

2. Please change "female” or "male" to "woman” or "man" as appropriate, when used as a noun (see for instance https://apastyle.apa.org/style-grammar-guidelines/bias-free-language/gender).

Response: Thank you for your comment. Done.

3. Please note that according to our submission guidelines (http://journals.plos.org/plosone/s/submission-guidelines), outmoded terms and potentially stigmatizing labels should be changed to more current, acceptable terminology. For example: “Caucasian” or "European Caucasoid" should be changed to “white” or “of [Western] European descent” (as appropriate).

Response: Thank you for your comment. Done

4. One of the noted authors is a group; OSA Group.

In addition to naming the author group, please list the individual authors and affiliations within this group in the acknowledgments section of your manuscript.

Please also indicate clearly a lead author for this group along with a contact email address.

Response: Thank you for your comment. Done

5. Please include captions for your Supporting Information files at the end of your manuscript, and update any in-text citations to match accordingly. Please see our Supporting Information guidelines for more information: http://journals.plos.org/plosone/s/supporting-information

Response: Thank you for your comment. Done

REVIEWER 1

In table two the term mulatto is used, this is considered to be a derogatory term in English and consideration should be given to replacing it with “Mixed race” or similar.

Response: Thank you for your comment. Done

The sentence Similarly, the working position of both parents accounted for a majority of permanent job positions (mothers: 56.7%; fathers: 57.2%) (Table 2).” Is somewhat unclear could this be presented as an amalgamated % of employment and then described or reworded “ The majority of parents were in fulltime employment (56.7% of mothers and 57.2% of fathers) . The information in table two is not aligned in Table 2 this causes some issues of clarity.

Response: Thank you for your comment. We rephrased the sentence following your advices and we fixed the tables format and numbers.

In the limitations section “helps the grown sons to reach a similar social and occupational status” should be reworded to a gender neutral term.

Response: Thank you for your comment. We reworded it to ensure gender neutrality.

REVIEWER 2

the authors declare no conflict of interests but some of them are affiliated with some of the listed Italian schools: could this have influenced how data was handled? could this have influenced the choice for some institutions to take part or not (a third of all institutions fitted the inclusion criteria)?

Response: Thank you for your comment. The data have not been stratified per school, so it was not possible (eventually) to influence the data handling (and we ensure that no attempt was even made). We are not aware about the reasons why some institutions decided not to participate since the form just asked if they were interested in participating or not. The inclusion criteria were at least one of the EN16686 criteria: as many as 4,800 hours of education, as many as 240 European Credit Transfer System (ECTS) with at least 60 ECTS of European Qualifications Framework (EQF) level 7 (EQF level 7 degree). Those data were first controlled on the schools websites and when weren’t found there, specific requests were made to the school via mail. 49 out of 61 schools fitted the inclusion criteria (80.33%), then 22 out of 49 accepted to participate.

I would suggest considering changing the title and description in the manuscript of the population assessed: they are referred as Italian students but it may be more accurate to describe them as students in Italy. First because some Italian study abroad (e.g. in the UK) and secondly because you may have non-Italian students in your sample (although this is unclear as I don't think you report the answer to question 12 of the questionnaire regarding their nationality)

Response: Thank you for your comment. We rephrased as “osteopathy students in Italy”. Moreover, we missed translating the “foreigner” line in the Table 3. Now It’s visible the number of foreigners.

some of the questionnaire questions have "other" as a possible answer. Did they lead to an open-ended box where they could enter some text? If so how were the data handled? This is not detailed in the methods or results sections clearly.

Response: Thank you for your comment. The “other” answers were closed ended, so, no further data are available other than the “other” answer itself. We added a specification of “closed ended items” in the methods section (2.3).

how was the questionnaire set: could participants select several answers or only one? e.g. for question 24 on school choice (and the results on that question are not presented in a table - if too many tables are required maybe they could be added in supplementary material?)

Response: Thank you for your comment. The survey allowed participants to respond using only a single answer or multiple answers depending on the question itself. Indeed, some % sums exceed the 100%, those questions were those where it was possible to submit multiple answers. Following your suggestions we provided all the supplementary tables whom data does not appear in the manuscript (S1 Tables).

From here in-text comments:

To be more accurate, please specify that this is for non-USA countries.

Response: Thank you for your comment. We removed “recently” to improve the accuracy of the sentence.

Whilst some osteopathic educational models are delivered in universities (e.g. in the UK, Switzerland, Australia etc), some countries with legal regulations do not share this (e.g. in France) so regulation does not de facto lead to training delivered in universities.

Response: Thank you for your comment. We added a specification following your kind comment.

health care provider?

Response: Thank you for your comment. Done.

As discussed above, this is not always an automatic switch - if this is automatic in Italian context, it would be usefl to provide exmaples and evidence of this.

Response: Thank you for your comment. We added a specification to clarify the concept.

The students of number expected to praticipate is different from the usual sampling frame. Whilst the number of participants in this study fulfill both numbers, I would suggest to the authors to mention the number of participants needed for a 95 percent

confidence level with an accepted margin of error of 3 (or acceptably 5) percent, e.g. see Sue, Valerie M., and Lois A. Ritter. Conducting online surveys. Sage, 2012.

Response: Thank you for your comment. We rephrased the paragraph as follow “The sample size was calculated based on information obtained through the web and Italian Association of Schools of Osteopathy, which supported an estimated sample of 7,000 students. According to the literature, it is assumed that the response rate ranged between 50% to 60% of those receiving the questionnaire [12]. In our survey we supposed that the percentage of non-respondents was 50% of OEIs students, so at 95% of the confidence level with a 5% margin of error, the percentage of the students was expected to range between 45% and 55% with a number ranging from 3,150 to 3,850 [26]. Sue, Valerie M., and Lois A. Ritter. Conducting online surveys. Sage, 2012.”

The percentage column is not easy to understand in the current format

Response: Thank you for your comment. Reformatted.

It is unclear what abroad stands for - does it mean outside Italy or in a different italian region. Abroad should mean outside Italy but it's unclear how those students would have been recruited as only Italian OEIs were invited to join. If abroad means region of studying being different forn the region of residence, please rephrase as it's a misuse of the word.

Response: Thank you for your comment. We rephrased as follow “offsite from their place of residence”

The questionnaire items for health status are rather imprecise. Students experience of university life includes depression, anxiety and burnouts. Whilst this could be encapsulated in CNS impaired system, it's currently hard to get a sense of their well-being with the broad categories used. I would suggest discussing this for further research opportunities.

Response: Thank you for your comment. We totally agree with your point, since the response “other” was closed ended we can’t exclude that psychological disorders may have fallen into that answer. Therefore we added the following sentence: “Noticeably, 244 students, accounting for 6.5% of the participants, declared to suffer from “other” health issues. It is well known that anxiety, depression and burnout are frequent conditions among students. Since the answer “other” was closed ended it is not possible to determine to what extent respondents included psychological disorders into that category. It would be interesting for future studies to further investigate the prevalence of psychological conditions among osteopathy students.”

please change to None

Response: Thank you for your comment. Done.

This sentence is unclear - what is there a need for an even distribution if patients don't favour one gender?

Response: Thank you for your comment. We agree, we opted for removing that confusing statement.

I initially was not sure what the link with the manuscript data was but the next sentence with nursing is clearer. The bridge needs to be reconsidered. There may be a leap of faith here with Christianity being the only religion in Italy. Arguably practitioners have to be trained and equiped to help patients from all faiths/spritual backgrounds.

Response: Thank you for your comment. We added the following sentence to further clarify the concept “However, an excessive polarization towards a single religion (as well as towards every cultural trait) might expose students to a lack of “diversity” awareness, resulting in a lack of multicultural care competence. Therefore, it might be needed to incorporate multicultural care training into the osteopathic curriculum to allow students to help patients from all faiths/spiritual backgrounds”

The link between PPE and spirituality is unclear - please explore/detail further

Response: Thank you for your comment. We rephrased the sentence to make it clearer. However it is grounded on a previous paper from Consorti et al. which studied the degree of acceptability of PPE among medical and osteopathy students. The results showed that religious factors weren’t associated with the degree of acceptability in either of the groups.

surveyed (not interviewed)

Response: Thank you for your comment. Done

is rarely considered (there are some so it can't say it's not considered)

Response: Thank you for your comment. Done

Self-employed?

Response: Thank you for your comment. Done.

not "the"

Response: Thank you for your comment. Done.

Main?

Response: Thank you for your comment. Done-

musculoskeletal pain?

Response: Thank you for your comment. Done.

redundant: osteopath? (remove professional?)

Response: Thank you for your comment. Done.

unclear what tandem refers to here

Response: Thank you for your comment. Re-worded to “team”.

This reference could be useful too as regarding osteopathic students (though in US) 

Lapinski, Jessica, et al. "Factors modifying burnout in osteopathic medical students." Academic Psychiatry 40.1 (2016): 55-62.

Response: Thank you for your comment. Added.

Do you mean as defined in introduction regarding the two types of osteopathy (US vs rest of the word) or osteopathy as practised in Italy?

Response: Thank you for your comment. We rephrased to make it clearer. Actually “medicine” was confusing.

Whilst these questions are interested there are not related to the topic as they refer to newly graduates (compared to current students). A similar project was conducted in the UK in 2012, "New Graduates’ Preparedness to Practise" commissioned by the General Osteopathic Council, conducted by Prof Della Freeth, Dr Paul McIntosh and Dr Dawn Carnes

Response: Thank you for your comment. We added a sentence at the beginning specifying what follows “In order to assess the outcome of the curriculum in future studies”

patients' needs or the needs of the patients

Response: Thank you for your comment. Done

This statement is confusing as osteopathic medical students are specific to the USA. Would possibly be more appropriate to separate medical osteopaths to non-medical osteopaths? So that your data from non-medical osteopaths could be more accurately compared to non-Italian but similar context

Response: Thank you for your comment. We rephrased deleting “medical” since the statement was related both to US and non-US students as reported afterwards.

I'm not sure that the number of students in the UK increases as the number of institutions is decreasing. Evidence for this statement is required.

Response: Thank you for your comment. We provided data regarding the registered numbers of osteopaths as a surrogate measure of previously enrolled students specifying that “However, Australian and UK data refers to professionals. Therefore it has to be considered a delay of, at least, 4 years from the time of their enrollment into the osteopathic school”. 

This sentence needs rewriting

Response: Thank you for your comment. Done as follow “On the other hand, ethnic minorities are under-represented among osteopathy students in Italy”.

The logic in this sentence is unclear - how student profile will help policymakers make decision regarding cnsultations and referrals?

Response: Thank you for your comment. It actually doesn’t. We rewrote it as follow “in order to assist policy-makers and educational institutions to tailor their offer to this specific target”

Limitations

Response: Thank you for your comment. Done

For any further information, please do not hesitate to contact us.

---

## [Editor Report · Decision Letter 1]

8 Feb 2021

Osteopathy students profile in Italy: a cross sectional census

PONE-D-20-33047R1

Dear Dr. Persiani,

We’re pleased to inform you that your manuscript has been judged scientifically suitable for publication and will be formally accepted for publication once it meets all outstanding technical requirements.

Kind regards,

Jenny Wilkinson, PhD

Academic Editor

PLOS ONE

Additional Editor Comments (optional):

Thank you for addressing the reviewer comments, the comments and revisions have satisfactorily addressed the reviewer comments

---

## [Editor Report · Acceptance letter]

10 Feb 2021

PONE-D-20-33047R1 

Osteopathy students profile in Italy: a cross sectional census 

Dear Dr. Persiani:

I'm pleased to inform you that your manuscript has been deemed suitable for publication in PLOS ONE. Congratulations! Your manuscript is now with our production department. 

Kind regards, 

on behalf of

Dr Jenny Wilkinson 

Academic Editor

PLOS ONE